# Inulin Supplementation Disturbs Hepatic Cholesterol and Bile Acid Metabolism Independent from Housing Temperature

**DOI:** 10.3390/nu12103200

**Published:** 2020-10-20

**Authors:** Mira J. Pauly, Julia K. Rohde, Clara John, Ioannis Evangelakos, Anja Christina Koop, Paul Pertzborn, Klaus Tödter, Ludger Scheja, Joerg Heeren, Anna Worthmann

**Affiliations:** 1Department of Biochemistry and Molecular Cell Biology, University Medical Center Hamburg-Eppendorf, 20246 Hamburg, Germany; m.pauly@uke.de (M.J.P.); juliarohde@online.de (J.K.R.); c.john@uke.de (C.J.); i.evangelakos@uke.de (I.E.); paulpertzborn@gmx.de (P.P.); toedter@uke.uni-hamburg.de (K.T.); l.scheja@uke.de (L.S.); heeren@uke.de (J.H.); 2Department of Internal Medicine I, University Medical Center Hamburg-Eppendorf, 20246 Hamburg, Germany; a.koop@uke.de

**Keywords:** fiber, inulin, short chain fatty acids, bile acids, cholestasis

## Abstract

Dietary fibers are fermented by gut bacteria into the major short chain fatty acids (SCFAs) acetate, propionate, and butyrate. Generally, fiber-rich diets are believed to improve metabolic health. However, recent studies suggest that long-term supplementation with fibers causes changes in hepatic bile acid metabolism, hepatocyte damage, and hepatocellular cancer in dysbiotic mice. Alterations in hepatic bile acid metabolism have also been reported after cold-induced activation of brown adipose tissue. Here, we aim to investigate the effects of short-term dietary inulin supplementation on liver cholesterol and bile acid metabolism in control and cold housed specific pathogen free wild type (WT) mice. We found that short-term inulin feeding lowered plasma cholesterol levels and provoked cholestasis and mild liver damage in WT mice. Of note, inulin feeding caused marked perturbations in bile acid metabolism, which were aggravated by cold treatment. Our studies indicate that even relatively short periods of inulin consumption in mice with an intact gut microbiome have detrimental effects on liver metabolism and function.

## 1. Introduction

Obesity and associated disorders have become a major global health burden, and preventive measures as well as effective treatments are strongly required. Obesity arises as a consequence of imbalanced energy intake and energy expenditure, and interventional approaches targeting one or the other have been introduced. One way to increase energy expenditure is activating the brown adipose tissue (BAT). In a process termed adaptive thermogenesis, uncoupling of the respiratory chain in BAT expends high amounts of energy and results in heat production. Of note, activation of BAT lowers plasma glucose and triglyceride levels and reduces insulin resistance [1,2,3]. Furthermore, activation of BAT by its natural stimulus cold, promotes cholesterol elimination via triggering hepatic remnant lipoprotein uptake and increasing hepatic bile acid synthesis, as well as fecal bile acid excretion. Overall, these processes protect against the development of atherosclerosis [4,5]. Strategies to reduce energy intake, on the other hand, mainly involve dietary modifications. Here, one appealing approach is to enrich diets with plant fibers, which are not only generally considered to be healthy but also convey a feeling of satiety. Fibers cannot be broken down by human digestive enzymes, but are fermented by the gut bacteria into the major short chain fatty acids (SCFAs) acetate, propionate, and butyrate. Inulin is a naturally occurring soluble fiber that is emerging as a food supplement and additive in highly processed foods to improve nutritional value. In rodents, inulin has been shown to lower plasma cholesterol and triglyceride levels [6], reduce postprandial hypertriglyceridemia [7], suppress adiposity [8], and protect against metabolic syndrome [9]. In addition, data from human studies suggest that dietary inulin supplementation might be able to reduce body weight [10,11]. To date, it is under debate whether inulin mediates these effects directly via its impact on the gut bacteria and subsequent interleukin-22 production [9] or indirectly via production of SCFAs and action on their corresponding receptors, GPR41 and/or GPR43 [12]. Of note, numerous studies have highlighted the role of SCFAs for regulation of appetite [13,14], development of obesity [15,16] and fatty liver [17], as well as insulin sensitivity and energy expenditure [18].

However, in recent years, several studies have emerged questioning the beneficial impact of fiber supplementation on (metabolic) health. For instance, Miles et al. [19] demonstrated that inulin supplementation aggravates colitis, and Hoving et al. [20] observed exacerbated atherosclerosis in response to inulin supplementation. Moreover, Janssen et al. [21] found that the highly fermentable dietary fiber guar gum enhanced hepatic inflammation and fibrosis. In the same line, a recent study published by Singh et al. [22] demonstrated that long-term inulin supplementation caused hepatocellular carcinoma in dysbiotic toll-like receptor 5 (*Tlr5*) knockout (KO) mice. Further, they noted that inulin caused bilirubinemia and liver inflammation. In line with previous observations by other groups [21,23,24], they also detected alterations in hepatic bile acid metabolism and serum bile acid levels after long-term inulin supplementation. While inulin-induced cancer formation was partly diminished after depletion of the gut bacteria by treatment with the antibiotic compound vancomycin, the inulin-dependent alterations in bile acid metabolism and associated cholemia were unaffected [25]. However, the underlying mechanisms remain obscure.

In the present study, we aim to investigate the effects of short-term inulin supplementation on hepatic cholesterol and bile acid metabolism in normobiotic wild type (WT) mice. We hypothesized that potential inulin-associated changes in cholesterol and bile acid metabolism might be attenuated by BAT activation. For this purpose, we studied WT mice, which were either housed under control or under cold conditions to active BAT.

We found that a short-term inulin supplementation altered systemic cholesterol and bile acid metabolism. While plasma cholesterol levels and fecal cholesterol excretion were diminished in response to inulin treatment, inulin supplementation caused marked perturbations in hepatic cholesterol metabolism, which were accompanied by cholemia as characterized by increased plasma bile acid and bilirubin levels. Of note, cholemia was even aggravated after cold housing. Overall, our results suggest that short-term inulin supplementation may disturb hepatic cholesterol and bile acid metabolism.

## 2. Materials and Methods

### 2.1. Experimental Animals, Housing Conditions, Diets, and Animal Experiment

All animal experiments were conducted in accordance with FELASA guidelines and approved by the Animal Welfare Officers of the University Medical Center Hamburg-Eppendorf (UKE), as well as the Behörde für Gesundheit und Verbraucherschutz Hamburg (animal protocol 15/96, approved October 08, 2015). Eleven to twelve week old male C57BL/6J mice were purchased from Janvier Laboratories. All mice were initially housed at 22 °C, with a day and night cycle of 12 h and ad libitum access to food and water. At the start of the experiment (day 0), mice were randomized based on body weight into four cohorts. The two ICD (inulin containing diet) cohorts received a diet containing 30 percent inulin (Ssniff S5714-E716), and the two CD (control diet) cohorts were fed a respective control diet, where the proportion of inulin was replaced with starch and maltodextrin (Ssniff S5714-E710) throughout the experiment. For detailed information on diet composition, see Appendix A. Five days after starting the dietary intervention (day 5), mice were exposed to either thermoneutral (30 °C) or cold (6 °C) housing conditions for seven days until the organ harvest (day 12). The experimental setup is depicted in Figure 1A. During this short-term experimental period, mice were housed in single cages to avoid group cuddling and warming during the cold housing intervention. Food intake was measured every two to three days, and body weight was monitored at least weekly. Mice were anesthetized after a 4 h fasting and 4 h refeeding period with a lethal dose of ketamine and xylazine. Cardiac blood was drawn with syringes containing 5 μL 0.5 M of EDTA. Animals were perfused with PBS containing 10 U/mL of heparin, then the organs were harvested and immediately stored at −80 °C or conserved in 3.7% formaldehyde solution for histology.

### 2.2. Plasma Analysis

Plasma cholesterol and triglycerides were determined using a commercial kit (Roche, Mannheim, Germany). Precipath^®^ (Roche, Mannheim, Germany) was used as a standard for both cholesterol and triglycerides. Total bilirubin was determined using a kit (Randox Labaratories, Crumlin, UK) according to the manufacturer’s instructions but adapted to a 96-well plate. Bilirubin (Sigma-Aldrich, St. Louis, MO, USA) was used as a standard.

Alanine transaminase (ALT) activity was determined using a commercial kit (Sigma Aldrich, St. Louis, MO, USA) according to the manufacturer’s instructions. One unit of ALT was defined as the amount of enzyme that generates 1.0 µmole of pyruvate per minute. The generated pyruvate was quantified calorimetrically every 5 min until the final measurement after 25 min.

### 2.3. Quantification of Fecal Cholesterol

Fecal samples were dissolved in methanol containing butylated hydroxytoluene (BHT) (1 mg/mL), using the TissueLyzer (Qiagen, Hilden, Germany). The methanol was evaporated in a vacuum centrifuge, and the dried pellets were re-dissolved in a buffer (0.1 M of Tris-HCl, pH = 7.5) before cholesterol was quantified by a colorimetric assay (Roche, Mannheim, Germany).

### 2.4. Liver Lipids

Liver tissue samples were homogenized in RIPA buffer using the TissueLyzer (Qiagen, Hilden, Germany) and centrifuged (13000 ×*g*, 5 min). Supernatants including the fat layer were harvested, and triglycerides were determined in the lysates diluted with PBS using a colorimetric assay (Diasys, Waterbury, CT, USA). Liver cholesterol levels were determined using Amplex^®^ Red Cholesterol Assay Kit (Thermo Fisher Scientific, Waltham, MA, USA) as described previously [26]. Protein concentrations were measured by the method of Lowry [27], and lipid concentrations are presented as μg/mg protein.

### 2.5. Gene Expression Analysis

Tissues were disrupted in TRIzol (Ambion, Life Technologies, Carlsbad, CA, USA) using a TissueLyzser (Qiagen, Hilden, Germany). Nucleic acids were extracted with chloroform, and total RNA was isolated using the RNA purification kit NucleoSpin^®^RNA II (Macherey & Nagel, Düren, Germany). Reverse transcription was performed by using the high capacity cDNA reverse transcription kit (Applied Biosystems, Thermo Fischer Scientific, Waltham, MA, USA). Quantitative real-time PCR was conducted on an ABI 7900HT detection system using TaqMan on-demand primer sets (Thermo Fisher Scientific, Waltham, MA, USA, *Abcb11*: Mm00445168_m1, *Abcc3*: Mm00551550_m1, *Abcc4*: Mm01226381_m1, *Abcg5*: Mm00446249_m1, *Abcg8*: Mm00445970_m1, *Baat*: Mm00476075_m1, *Ccl2*: Mm00441242_m1, *Col1a1*: Mm00801666_g1, *Cxcl1*: Mm00433859_m1, *Cyp27a1*: Mm00470430_m1, *Cyp7a1*: Mm00484150_m1, *Cyp7b1*: Mm00484157_m1, *Cyp8b1*: Mm00501637_s1, *Fgf15*: Mm00433278_m1, *Fxr*: Mm00436419_m1, *Hmgcr*: Mm01282499_m1, *Il1b*: Mm00434228_m1, *Il6*: Mm00446190_m1, *Ldlr*: Mm00440169_m1, *Lrh1*: Mm00446088_m1, *Lrp1*: Mm00464608_m1, *Nr0b2*: Mm00442278_m1, *Slc10a1*: Mm00441421_m1, *Slc51b*: Mm01175040_m1, *Slco1b2*: Mm00451510_m1, *Srebpf1*: Mm00550338_m1, *Srebpf2*: Mm01306292_m1, *Sult2a1*: Mm04205657_mH, *Timp1*: Mm00441818_m1, *Tnf*: Mm00443258_m1). Levels of mRNA were normalized to the level of the housekeeping gene TATA box binding protein (*Tbp*) mRNA. Results were displayed as a relative expression normalized to the experimental control group.

### 2.6. Bile Acid Measurement

Targeted bile acid analysis was performed by HPLC coupled to electrospray ionization tandem mass spectrometry [28]. Briefly, sample preparation comprised a simple methanol liquid-liquid extraction of tissues and bio fluids. Quantitative measurement of bile acids was performed using a LC-ESI-QqQ system run multiple reaction monitoring (MRM) mode: HPLC: NEXERA X2 LC-30AD HPLC PUMP (Shimadzu, Tokyo, Japan); column: Kinetex C18 (100 Å, 150 mm × 2.1 mm i.d., Phenomenex, Torrance, CA, USA); QqQ: Q trap 5500 System (SCIEX, Darmstadt, Germany). Peak identification and quantification was performed by comparing retention times as well as MRM transitions and peak areas, respectively, to particular corresponding standard chromatograms.

### 2.7. Histology

Liver tissues were fixed in formalin (3.7%) and embedded in paraffin. Sections were stained with hematoxylin and eosin (H&E) and Sirius Red for fibrillary collagen using standard protocols. Sirius Red staining was quantified in at least five low power fields using Adobe Photoshop and ImageJ.

### 2.8. GC-FID Based Analysis of SCFAs

Thirty milligrams of cecal content or 30 µL of plasma were extracted in 295.5 μL ethanol, and iso-butyric acid was added as an internal standard. The samples were homogenized using a Tissue Lyzer (Qiagen, Hilden, Germany) if necessary and then centrifuged (10 min, 13,000× *g*). Five microliters of 0.8 M of NaOH were added to the supernatant, and it was evaporated using a vacuum centrifuge. The residual salts were re-dissolved in 50 μL of EtOH and 10 μL of 0.6 M of succinic acid. Samples were separated by a gas chromatograph (Hewlett Packard 5890 Series II) equipped with a Nukol Fused Silica Capillary Column (15 m × 0.32 mm × 0.25 μm film thickness) and helium as a carrier gas. The oven started with an initial temperature of 70 °C and heated up at 30 °C/min until reaching 100 °C. Then, the oven continued heating up at 6 °C/min until reaching the final temperature of 190 °C. SCFAs were detected with a flame ionization detector. Peaks were identified by comparing retention times and peak areas to standard chromatograms.

### 2.9. Satistical Analysis

Data are expressed as mean ± S.E.M. Statistical analysis was performed using GraphPad Prism 7.0. Group sizes were chosen based on the key parameter plasma cholesterol, assuming an effect size (25%) and standard deviation (15%) observed in previous studies. Two-way ANOVA was conducted using log-transformed data to achieve a normal distribution. Differences were considered as significant at a probability level (*p*) of 0.05. Same letters designate groups that are not significantly different from each other.

## 3. Results

### 3.1. Short-Term Dietary Inulin Supplementation Does Not Affect Food Intake and Body Weight Gain

While long-term inulin feeding has been shown to result in disturbed liver bile acid metabolism, cholestasis, liver damage, and even hepatocellular cancer [22], our present study aimed to investigate the impact of short-term (12 days) dietary inulin supplementation. Further, we speculated that BAT activation by cold exposure might affect cholestasis development, as it is known to alter the expression of bile acid synthesis enzymes and promote fecal excretion of bile acids [4]. For this purpose, we fed WT mice either with a control diet (CD) or an inulin containing diet (ICD). Initially, all mice were housed at 22 °C, but after five days they were exposed to warm control (30 °C) or cold conditions (6 °C) until right before sacrifice, as depicted in Figure 1A. As expected, both CD- and ICD-fed cold housed mice increased their daily food intake and ate roughly 1.5 g more than their warm housed counterparts. Of note, inulin supplementation did not affect daily food intake (Figure 1B). While initially, CD- and ICD-fed mice lost some body weight, during the course of the study, all mice started to gain weight again, and food intake, body weight, and liver weight were not affected by inulin supplementation (Figure 1C,D). Interestingly, in ICD-fed mice, cold housing caused an increase in liver weight (Figure 1D).

### 3.2. Short-Term Dietary Inulin Supplementation Leads to an Increased Production of SCFAs

Inulin is fermented into SCFAs by gut bacteria. In mice, bacterial density is highest in the cecum, and in line with this it is considered as the principal organ for fermentation. In order to verify successful inulin supplementation, we analyzed cecum weights and SCFA levels in CD- and ICD-fed warm or cold housed animals. As expected, cecum weights were higher after inulin supplementation irrespective of housing conditions (Figure 2A), while cold housing alone did not affect cecum weights (Figure 2A). Total and individual (acetate, propionate, and butyrate) cecal SCFA levels were higher in both warm and cold housed ICD-fed mice (Figure 2B). Of note, cecal SCFA levels were also higher in cold housed CD-fed mice compared to the warm housed CD mice (Figure 2B), but not as high as after inulin supplementation.

SCFAs are quickly absorbed into colonocytes, where they serve as an energy substrate or might activate their receptors GPR41 and GPR43. Neither inulin supplementation nor cold housing affected the colonic expression of GPR41 (Appendix A), while GPR43 expression remained similar after inulin supplementation, but was lower in cold housed mice, which received the control diet (Appendix A). Alternatively, SCFAs are drained into the portal vein and eventually reach the liver. Here, most of the SCFAs are taken up by hepatocytes, whereas only a small proportion reaches systemic circulation. To assess how inulin supplementation affects exposition of the liver and peripheral organs to SCFAs, we analyzed their concentrations in portal and systemic plasma. While we were still able to detect higher amounts of acetate, propionate, butyrate, and total SCFAs in the portal circulation after inulin supplementation (Figure 2C), only propionate and butyrate levels were elevated in the systemic circulation of both warm and cold housed ICD-fed mice (Figure 2D). Interestingly, we detected moderately higher levels of acetate, propionate, and total SCFAs in portal but not in systemic circulation of cold housed mice fed the control diet. There might be an additive effect of cold housing and inulin supplementation, as levels of butyrate and total SCFA levels were significantly higher compared to all other groups (Figure 2C). Overall, inulin supplementation resulted in higher SCFA production and higher SCFA levels in the portal and even in the systemic circulation.

### 3.3. Effects of Short-Term Inulin Supplementation on Cholesterol and Bile Acid Metabolism

There have been conflicting reports about the impact of inulin supplementation on systemic cholesterol levels and cholesterol metabolism [6,23], and most importantly, studies investigating the combined effect of inulin supplementation and BAT activation have not been carried out so far. To address these questions, we measured plasma lipid levels in the warm and cold housed CD- and ICD-fed mice. In line with previous reports [2,4], cold treatment resulted in profoundly lower plasma triglyceride and cholesterol levels irrespective of ICD feeding (Figure 3A). Interestingly, while inulin supplementation did not affect plasma triglyceride levels, plasma cholesterol levels were decisively lower in both ICD-fed groups. Of note, inulin lowered cholesterol levels to a similar extent as cold exposure. However, we did not observe an additive effect of ICD feeding and cold housing (Figure 3A).

To study if reduced plasma cholesterol levels stem from impaired intestinal cholesterol uptake or enhanced cholesterol excretion, we measured fecal cholesterol levels. While fecal cholesterol levels were similar in warm and cold housed CD-fed mice, inulin supplementation led to slightly lower fecal cholesterol levels in a warm housing environment (*p* = 0.0744) and even significantly lower cholesterol excretion after cold housing (Figure 3B). As the liver is the central organ for cholesterol homeostasis, we next analyzed hepatic cholesterol levels. In line with the reduced fecal excretion, cholesterol accumulated in the livers of ICD-fed mice. This effect was even more evident after combining ICD feeding and cold treatment (Figure 3C). The inulin-induced alterations in cholesterol excretion and hepatic cholesterol content prompted us to analyze the hepatic expression of genes relevant for cholesterol metabolism. To our surprise, inulin supplementation alone affected none of the genes analyzed (Figure 3D). In line with previous findings, cold housing lowered the expression of genes important for cholesterol synthesis and uptake (*Srebpf 1*, *Srebpf 2*, *Hmgcr*, *Ldlr*, *Lrh1*) and increased genes responsible for cholesterol disposal (*Abcg5*, *Abcg8*) compared to thermoneutral housing conditions. Strikingly, inulin supplementation seemed to reverse these cold-mediated effects (*Srebpf 2*, *Hmgcr*, *Ldlr*, *Abcg5*, and by trend also for *Abcg8* (*p* = 0.0979)).

Besides its direct biliary secretion, cholesterol can also be excreted after its conversion into bile acids. A dysregulated bile acid metabolism and highly increased plasma bile acid levels have been described after long-term inulin supplementation [29]. We therefore wanted to investigate if already short-term inulin supplementation causes dysregulated bile acid metabolism and if cold housing would ameliorate the inulin-induced perturbations. For this purpose, we measured bile acid levels in the portal and systemic circulation.

We found that short-term inulin supplementation moderately raised portal bile acid levels, mainly due to an increase solely in unconjugated bile acids, which were two-fold higher independent of housing conditions (Figure 4A). Of note, systemic levels of especially unconjugated bile acids increased dramatically after inulin supplementation in both warm and cold housed mice (Figure 4B). Higher systemic bile acid levels might result from higher hepatic synthesis of bile acids and/or decreased capacity of the liver to clear and secrete bile acids due to hepatocyte damage (cholestasis). We noted that inulin-fed mice displayed reduced fecal bile acid disposal compared to the control groups, possibly arguing for a reduced secretion of bile acids (Figure 4C). Interestingly, this was already evident in fecal samples collected after one week of inulin feeding and to our surprise was not influenced by housing conditions (Figure 4C). Next, we measured ileal levels of the FGF15 transcript, an intestinal hormone produced in response to bile acids. Strikingly, ileal expression of *Fgf15* was almost undetectable in ICD-fed mice compared to warm and cold housed control mice, indicating very low bile acid concentrations in ileal enterocytes (Figure 4D). Although not statistically significant, cold housing even intensified this effect (*p* = 0.0872) (Figure 4D). Physiologically, intestinal FGF15 acts on the liver and inhibits the expression of *Cyp7a1*, the rate limiting enzyme in classical bile acid synthesis, in a negative feedback loop [30]. Thus, we next analyzed hepatic expression of *Cyp7a1* and other bile acid synthesis genes. Surprisingly, except for a lower expression of *Cyp7b1*, the major enzyme mediating alternative bile acid synthesis, short-term ICD feeding under warm housing conditions did not affect the expression of other genes involved in bile acid synthesis. As described before [4], cold housing promoted the expression of bile acid synthesis genes (Figure 4E). Most strikingly and in line with the drastic reduction of ileal *Fgf15*, short-term inulin supplementation combined with cold treatment evoked a massive increase in *Cyp7a1*. Interestingly, we further noted that inulin supplementation impeded the cold-induced increases in *Cyp7b1* and *Cyp8b1* expression (Figure 4E). As *CYP7B1* derived bile acids were described to promote thermogenesis in BAT after cold treatment [4], we next investigated if inulin-induced reductions in *Cyp7b1* might affect BAT activation. Of note, despite the lower hepatic *Cyp7b1* expression in the cold ICD group, we did not detect altered expression of thermogenic genes *Ucp1*, Dio2, and *Ppargc1a* in BAT and WAT in response to inulin feeding (Appendix A). We next analyzed bile acid transporter genes. Inulin supplementation did not change the expression of the canalicular bile salt export pump BSEP (encoded by *Abcb11*), and differentially affected the expression of bile acid import pump NTCP (encoded by *Slc10a1*). However, we found that inulin supplementation led to strong increases in the expression of the basolateral bile acid export pumps MRP3 and MRP4 (encoded by *Abcc3* and *Abcc4)* (Figure 4F), as well as OSTβ (*Slc51b*) compared to the control group. Additionally, the expression of bile acid detoxifying *Sult2a1* was massively increased. Of note, all of these enzymes are implicated in the so-called alternative basolateral bile acid efflux, which occurs as a protective mechanism in response to hepatic bile acid overload [31]. This effect was even more pronounced in cold housed ICD-fed mice (Figure 4F). However, cold housing alone did not alter expression of bile acid transport genes (Figure 4F).

Together, these data support the notion that already short-term inulin supplementation results in disturbed cholesterol and bile acid metabolism, which is characterized by increased hepatic bile acid production through the major classical synthesis pathway and reduced fecal bile acid output. Furthermore, hepatic bile acid clearance and secretion are dysregulated, resulting in elevated serum bile acid levels. Against our initial hypothesis, cold housing could not ameliorate this effect, but rather exacerbated it.

### 3.4. Effects of Short-Term Inulin Supplementation on Liver Damage

As already short-term inulin supplementation caused marked perturbations in hepatic bile acid metabolism and indications of cholestasis, we wondered if this was accompanied by signs of liver damage. We therefore examined H&E-stained liver sections. In all groups, we did not detect obvious liver injuries. If any, we saw a tendency towards higher lipid deposition in both ICD-fed groups compared to CD-fed mice, and a lower lipid deposition in the cold housed group (Figure 5A). We could confirm this trend when analyzing liver triglyceride levels biochemically (warm CD vs. warm ICD, *p* = 0.073; cold CD vs. cold ICD, *p* = 0.324) (Figure 5B). As a measure of liver fibrosis, we next performed Sirius Red staining of liver sections. In agreement with the H&E sections, we did not detect serious signs of fibrosis in any group (Figure 5C,D). Given the high serum bile acid levels in our ICD-fed mice, we speculated that short-term inulin supplementation might also result in high bilirubin levels. We found that cold housing moderately increased plasma bilirubin levels compared to controls (Figure 5E). Of note, ICD supplementation resulted in higher plasma bilirubin levels in warm and even a little further in cold housed mice compared to CD-fed mice, although the results did not reach statistical significance (warm CD vs. warm ICD, *p* = 0.139; cold CD vs. cold ICD, *p* = 0.371) (Figure 5E). To further characterize potential liver damage, we next assessed plasma ALT levels. Except for slightly higher ALT levels in the warm housed ICD-fed mice (warm CD vs. warm ICD, *p* = 0.193), we did not observe any differences (Figure 5F). Finally, we studied gene expression of fibrosis and inflammation markers.

While we did not detect major differences in chemokine expression between the groups (*Cxcl1*, *Ccl2*, and *Cxcl10*), we found that cold housing resulted in markedly lower expression levels of inflammatory markers (*Il1b*, *Il6*, and *Tnfa*) in both CD- and ICD-fed mice. In line with the Sirius Red staining, the expression of *Col1a1* remained unaltered, but we detected a trend towards higher levels of the fibrosis marker *Timp1* after ICD feeding (warm CD vs. warm ICD, *p* = 0.063; cold CD vs. cold ICD *p* = 0.017) (Figure 5G). In summary, despite the dramatic changes in bile acid metabolism occurring after short-term inulin supplementation, we did not observe significant liver damage and only beginning signs of liver dysfunction.

## 4. Discussion

Many approaches to prevent or counteract the development of obesity involve dietary modifications to reduce energy intake and to improve nutritional value. During the last few years, dietary supplementation with plant derived fibers, which are generally considered as natural and non-hazardous, has gained great popularity. In particular, inulin has emerged as an appealing supplement, as several studies pointed out their beneficial metabolic effects [6,7,9]. However, more recent reports question this trend. Instead, they propose that plant derived supplements might have harmful side effects and cause hepatic fibrosis [21] and liver injuries [32] or, in the case of inulin, may even promote liver cancer [22]. Of note, the carcinogenic effect of inulin was described to occur after a rather long period of inulin supplementation, and in dysbiotic *Tlr5* KO mice, which are more susceptible to pathologies. However, the inulin-induced development of hepatocellular carcinoma was accompanied by cholestasis and major disturbances in hepatic bile acid metabolism, an effect described by others as well [23,25]. In the present study, we first aimed to investigate the effects of short-term (two weeks) inulin supplementation in normobiotic WT mice, with a special focus on hepatic cholesterol and bile acid metabolism. Second, we aimed to gain a deeper understanding of the mechanistic processes behind the alterations in bile acid metabolism. We hypothesized that dysregulated bile acid metabolism and cholemia might be ameliorated after activation of BAT by cold housing, as this has been shown to promote fecal bile acid excretion.

Our results indicate that short-term inulin supplementation in normobiotic mice does not affect body weight. Most likely, in contrast to human studies [10,11], the timeframe of supplementation was too short to reach any effect. Still, despite no observed differences in body weight, we were able to show that already short-term inulin supplementation dramatically reduced plasma cholesterol levels. Interestingly, while Trautwein et al. saw similar effects in hamsters [6], Hoving et al. and Mistry et al. reported unaltered plasma cholesterol levels in mice supplemented with inulin [20,23,24]. Of note, we found that these reductions were not caused by increased hepatobiliary cholesterol excretion or decreased intestinal cholesterol uptake, as indicated by hepatic gene expression and reduced fecal cholesterol levels. These findings are in agreement with a study of Hoving et al. showing that inulin supplementation is not able to ameliorate atherosclerosis [24]. In fact, our results indicate that reduced cholesterol levels stem from altered bile acid metabolism. In particular, we found that already short-term inulin supplementation, especially in combination with cold housing, raised circulating bile acid levels, and induced mild cholemia and hepatic bile acid synthesis. Most importantly, fecal bile acid deposition was diminished after inulin supplementation. In addition, the almost undetectable levels of *Fgf15* transcript in the intestine of ICD-fed mice suggest that hepatic bile acid excretion might be abrogated and that the low *Fgf15* copy numbers might result in lower FGF15 plasma levels, which in turn cause the deregulated high expression of *Cyp7a1*. In future studies, not only ileal *Fgf15* expression but also circulating FGF15 levels should be measured. Strikingly, as others before, we found that alternative bile acid detoxification and export routes were strongly increased by short-term inulin supplementation. Of note, these are mechanisms protecting hepatocytes from bile acid overload. Initially we hypothesized that cold treatment might ameliorate inulin-induced cholestasis, as cold treatment was shown to promote fecal bile acid disposal [4]. However, to our surprise, cold housing exacerbated this effect. One explanation would be that cold treatment not only promotes fecal bile acid excretion, but also increases hepatic bile acid synthesis via the induction of Cyp7b1. In turn, Cyp7b1-derived bile acids promote BAT activity [4]. Interestingly, although inulin feeding reverted the cold-stimulated induction of Cyp7b1, BAT activity and WAT browning were not affected by inulin treatment.

Although we detected marked perturbations in hepatic bile acid metabolism and cholestasis after inulin feeding, we only detected mild signs of liver damage and fibrosis in our model. Fecal excretion of bile acids was already strongly decreased after only one week of inulin supplementation, indicating that alterations in the canalicular bile acid secretion are rapid and therefore occur prior to liver damage. In the same line, a very recent study showed that the depletion of gram positive bacteria by vancomycin treatment [25] indeed reduced liver damage in inulin-supplemented mice, but failed to normalize bile acid metabolism. One explanation for the so far absent liver damage might be that in our model, serious hepatocyte damage is most likely prevented by compensatory increases in alternative bile acid export routes. Yet, inulin feeding mainly resulted in higher hepatic levels of liver toxic unconjugated bile acid species, which are known to promote liver cancer [25]. We assume inulin feeding might cause more severe liver damage and even liver cancer in our normobiotic WT in the long run. Thus, it is most likely that the time point of our analysis was too early to detect signs of severe liver damage.

One possible explanation for the observed dysregulated bile acid metabolism and the development of cholestasis might be alterations in the gut microbiota. Future studies should carefully evaluate alterations in the composition of the gut microbiota, as for instance, Singh et al. demonstrated that cholestasis is higher in dysbiotic *Tlr5-KO* mice and thus depends on the presence or absence of specific bacteria, which are more prone to generate toxic unconjugated bile acid species. Next, the gut bacteria could also mediate effects indirectly via products of bacterial fermentation, such as the SCFAs acetate, propionate, and butyrate. In response to inulin supplementation, we detected higher SCFA levels in the systemic circulation, which might eventually reach the liver and the peripheral organs. We speculate that SCFAs interfere with bile acid metabolism by signaling via their receptors GPR41, GPR43, and GPR109, which have been shown to impinge on lipid metabolism before [33]. However, in line with conflicting reports [34,35] on hepatic expression of GPR41 and GPR43, we were not able to detect transcripts of these receptors in the liver (data not shown) and only minor expression changes in the colon. Thus, further studies exploring the impact of SCFAs on BA metabolism via GPR41, GPR43, and GPR109 are warranted.

Overall, we were able to show that under certain circumstances, even short-term supplementation of inulin can disturb systemic cholesterol and bile acid metabolism, cause cholemia, and might promote liver damage in normobiotic mice. Neither alterations in the expression of canalicular bile acid transporters nor fibrotic changes in bile ducts or liver tissue seem to be responsible for the observed defects in canalicular bile acid secretion. As cold housing had an additive effect on the disturbances of hepatic bile acid metabolism, an increased hepatic synthesis of bile acids might be the cause of the observed changes. However, the underlying molecular mechanisms remain elusive. One potential drawback of our study might be the relatively high concentration of inulin used for supplementation. Still, as we saw dramatic changes in metabolism after a really short period, the development of inulin as a popular dietary supplement is worrying. Our study once more points out the potential risks and side effects of inulin, and that further research in this field is crucial.

## 5. Conclusions

Our studies indicate that even relatively short periods of inulin consumption in mice with an intact gut microbiome disturb systemic cholesterol and bile acid metabolism and may have detrimental effects on liver function.

## Figures and Tables

**Figure 1 nutrients-12-03200-f001:**
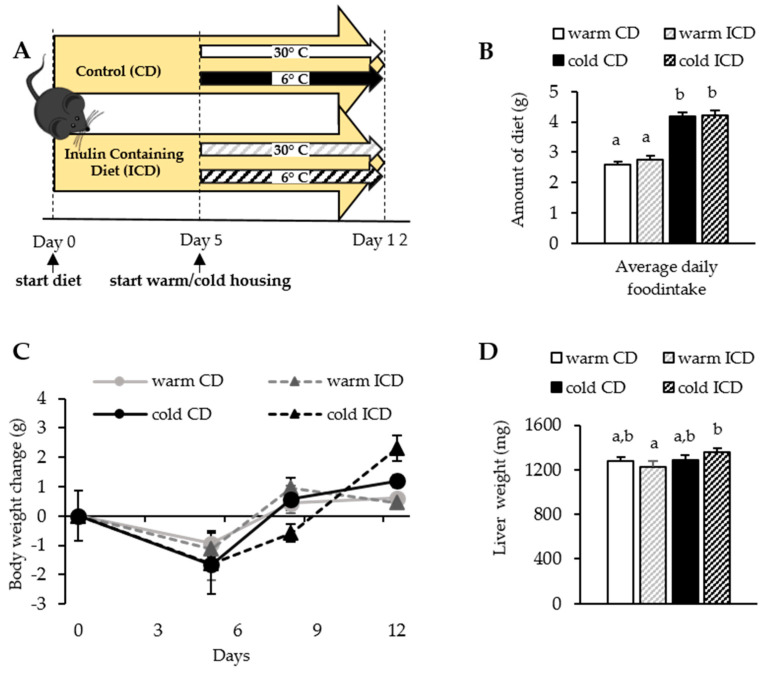
Short-term inulin supplementation does not alter body weight. (**A**) Study design: C57BL/6J mice were fed an inulin containing diet (ICD) or a respective control diet (CD) for 12 days, and were housed at 30 °C (warm) or 6 °C (cold) for seven days; (**B**) average daily food intake; (**C**) body weight change over time, and (**D**) liver weight after 12 days of feeding. Data are shown as mean values ± SEM; CD warm: *n* = 6; ICD warm: *n* = 6; CD cold: *n* = 6; ICD cold: *n* = 5; different letters indicate significant differences between groups (*p* < 0.05) determined by two-way ANOVA.

**Figure 2 nutrients-12-03200-f002:**
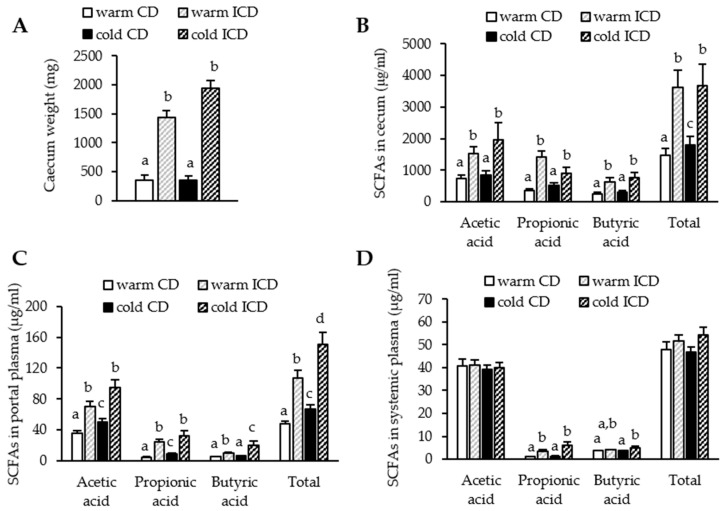
Inulin supplementation increases SCFA levels. (**A**) Cecum weight; SCFAs in (**B**) cecal content, (**C**) portal plasma and (**D**) systemic plasma. Data are shown as mean values ± SEM, and different letters indicate significant differences between groups (*p* < 0.05) determined by two-way ANOVA. CD warm: *n* = 6; ICD warm: *n* = 6; CD cold: *n* = 6; ICD cold: *n* = 5 (ICD cold cecal content *n* = 4). SCFAs: short chain fatty acids.

**Figure 3 nutrients-12-03200-f003:**
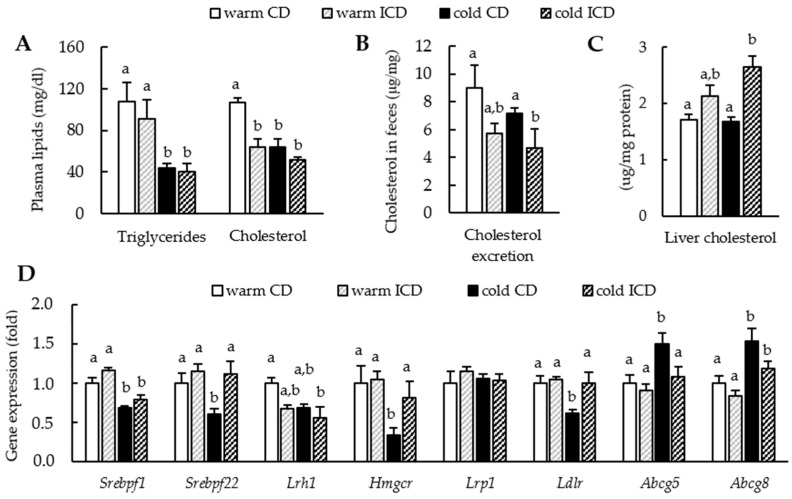
Inulin supplementation alters cholesterol handling. (**A**) Plasma lipids; (**B**) cholesterol levels in feces; (**C**) liver cholesterol levels and (**D**) relative expression of genes related to cholesterol metabolism normalized to *Tbp* as the housekeeper in livers. Data are shown as mean values ± SEM, and different letters indicate significant differences between groups (*p* < 0.05) determined by two-way ANOVA. CD warm: *n* = 6; ICD warm: *n* = 6; CD cold: *n* = 6; ICD cold: *n* = 5 (ICD cold plasma lipids *n* = 4).

**Figure 4 nutrients-12-03200-f004:**
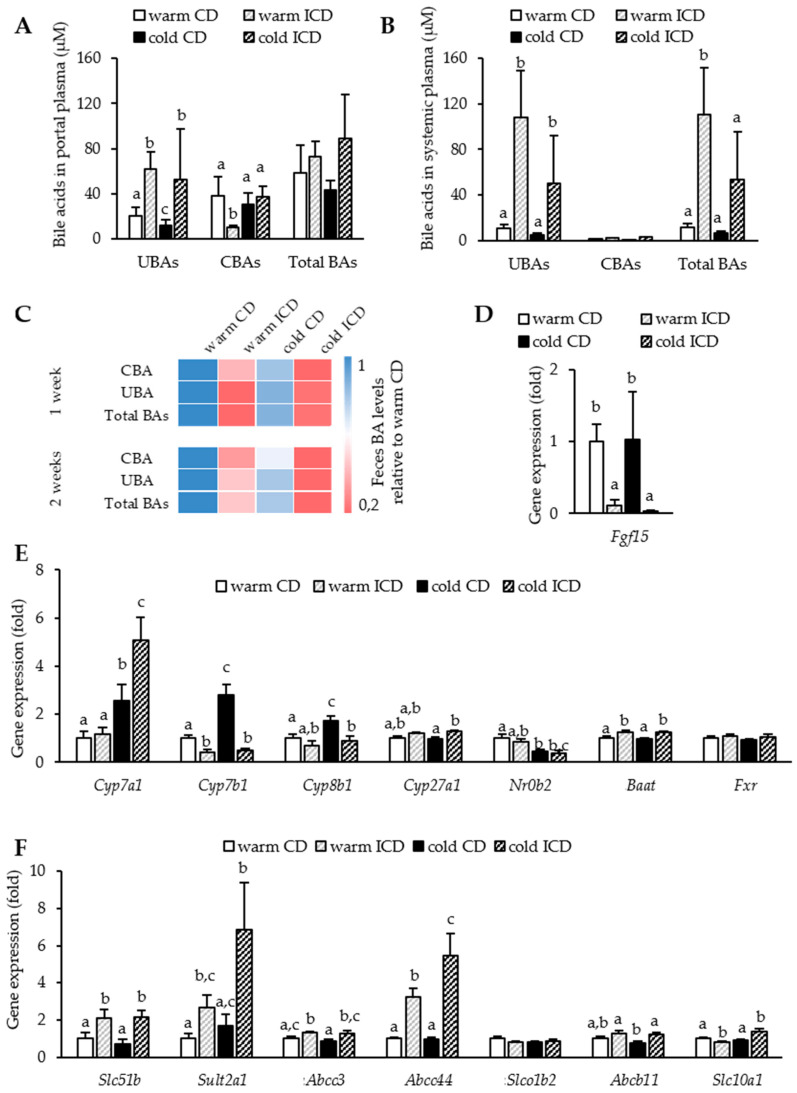
Inulin alters the bile acid metabolism. (**A**) UBA, CBA, and total bile acid levels in portal plasma and (**B)** systemic plasma, and in feces (**C**). Relative expression of (**D**) *Fgf15* in ilea; (**E**) genes involved in the synthesis of bile acids and its regulation in the liver. (**F**) Genes involved in the hepatic transport of bile acids. Levels are normalized to *Tbp* as the housekeeper. Data are shown as mean values ± SEM, and different letters indicate significant differences between groups (*p* < 0.05) determined by two-way ANOVA. CD warm: *n* = 6; ICD warm: *n* = 6 (ICD warm BA portal plasma *n* = 5); CD cold: *n* = 6 (CD cold BA systemic plasma *n* = 4); ICD cold: *n* = 5. VBA: urine bile acids, CBA: conjugated bile acids.

**Figure 5 nutrients-12-03200-f005:**
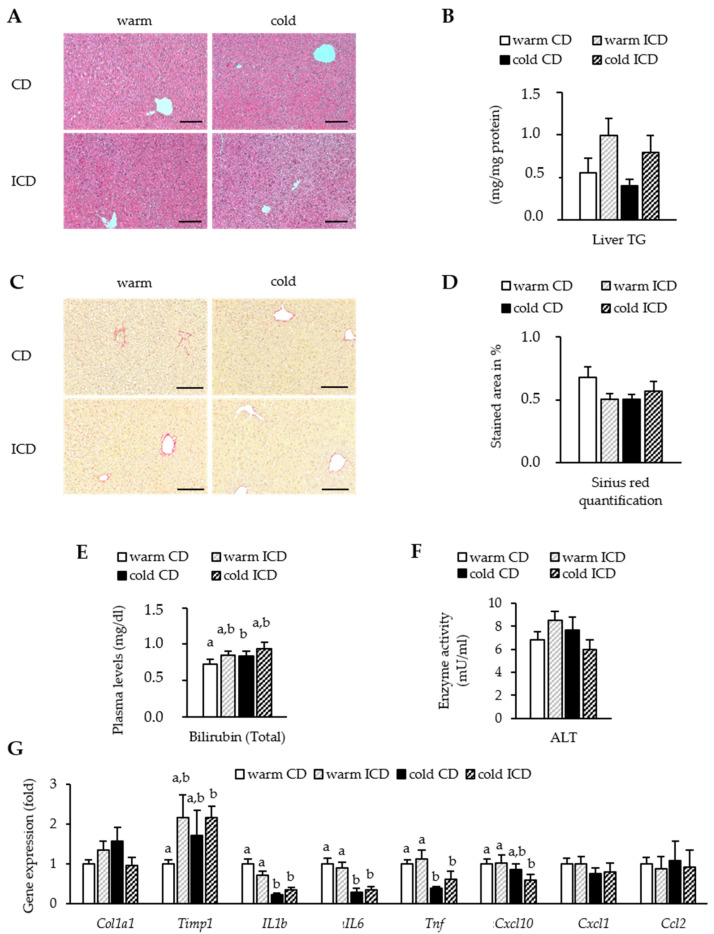
Inulin does not cause definite signs of liver damage. (**A**) H&E-stained (scale bar = 100 μm) sections of livers (**B**) and liver TG levels. (**C**) Sirius Red stained sections of livers (scale bar = 200 μm), including Sirius Red quantification (**D**). (**E**) Plasma levels of bilirubin; (**F**) plasma ALT enzyme activity; (**G**) relative hepatic expression of genes associated with liver damage, inflammation, or immune cell infiltration normalized to *Tbp* as the housekeeper. Data are shown as mean values ± SEM, and different letters indicate significant differences between groups (*p* < 0.05) determined by two-way ANOVA. CD warm: *n* = 6; ICD warm: *n* = 6 (ICD warm bilirubin and ALT *n* = 5); CD cold: *n* = 6 (CD cold bilirubin and ALT *n* = 5); ICD cold: *n* = 5.

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
