# Peer review of "Inulin Supplementation Disturbs Hepatic Cholesterol and Bile Acid Metabolism Independent from Housing Temperature"

_nutrients, 2020, doi:10.3390/nu12103200_

Round 1

Reviewer 1 Report

I read the article "Inulin supplementation disturbs hepatic cholesterol and bile acid metabolism independent from housing temperature" written by Pauly MJ et al.

The manuscript is well written in the sophisticated context.

 I am impressed with the experimental result that inulin worsened lipid accumulation in the liver in spite of decreased serum cholesterol level on cold inulin contained diet.  Cold stimulus has an effect of upregulating energy expenditure on the brown adipose tissue (BAT). Through the article, I think Inulin could offset the effect of upregulation of metabolism by activated BAT.

FGF15 is an interesting and unique peptide. That suppresses bile acids metabolism. I imagine that in the experiment, upregulated (or downregulated ) FGF15 levels in the ileum could affect  the bile acid metabolism the through the serum levels of FGF15. Is that true?  If you are prepared, you should measure the serum levels of FGF15 in each mice group. FGF15 is a crucial role in your whole experiment. Bile acid accelerates  activity of BAT through TGR5. FGF15 might also be a negative regulator for the BAT or positive one?  After all you might need the physiological and molecular biology data on the BAT.

Reviewer 2 Report

Introduction: final sentence (conclusion) in the introduction doesn't make much sense. I would suggest rewording it. 

Materials and Methods: the study design is not very clear. Was this a one time intervention? Please describe the protocol in more details and maybe add a figure/chart for clarity. Why did authors choose this one time feeding of inulin? Rationale needs to be stated in introduction with more details. 

  • Details of the diet needs to be described or add a table with the details. What is the equivalent of 30 percent of inulin or total fiber compared to human consumption?
  • Was gut microbiome done to compare or account for possible difference between groups? 
  • Not clear why mice were house individually, since these are social animals and usually housed in groups. 
  • Was there power calculations done for this study? Please mention or add to supplementary as it is not clear why such small n was used to an intervention with multiple factors (diet, temperature, housing).

Results: What is the n per group? Please add to text or figures. Not very clear on the legend.

  • Was the 5 days of temperature difference chosen based on preliminary data from the long term study? Please describe that. 
  • When describing the results, please mention compared to what group. For instance, the slight increase in liver weight for the ICD cold group was compared to the control or the ICR warm group? Was it 6 groups or 4? That is also not clear because from figure 1A it looks like 6 groups. Please add p-values when authors mention the word slightly for changes.
  • HIGHLY SUGGEST doing microbiome analysis if fecal samples were collected to further understand the changes in gut microbiota composition from beginning to end of intervention, as well as evaluate the effects of both inulin and temperature changes. 
  • Wording a bit confusing. The way authors are describing results, it seem like the intervention is affecting the same animal, but in fact it is between groups. To know if inulin or temperature reduced cholesterol, authors would have to measure before and after in the same mice. In this case it is comparing groups, so for this reason use lower or higher according to groups.
  • Excellent work done with the gene expression!

Discussion: Not very clear second paragraph in terms of the bacteria, temperature changes, and cancer. Please elaborate more. 

  • why were the SCFA receptors measured in the gene expression? There is a missing link between the inulin (fiber), SCFA, and bile acid metabolism based on the results presented. Please add more data if you have or rearrange results to make a clear picture. 
  • Author's conclusions are a bit confusing, because inulin supplementation does not seem to change expression of enzymes involved in bile acid metabolism, and rather the cold temperature. Therefore, the inulin (fiber) might not be the issue, but rather in the scenario of cold temperature, which in this case, probably carbohydrates and fat might be more beneficial due to BAT functionality. 
  • Author's started mentioned BAT, but not much is discussed about it. Please elaborate more based on results, or change the introduction accordingly. 

Round 2

Reviewer 2 Report

The authors have done an excellent job in addressing the comments/concerns previously provide. Figure 1A looks much better and it gives a clear picture of the study design. The discussion has been improved to a great extent which shows the understanding of the authors towards the topics and gaps in the literature. 

I would recommend adding to the results description on line 95-96 the reason why mice were house individually. As the authors have mentioned in the response, it is because of the study design to avoid group cuddling during the temperature phase of the experiment. Please mention this is is short term to show that authors acknowledge that mice are social animals. 

I would also highly recommend a power calculation based on previous studies you have published, to validate and strengthen the statistics of this study. It gives more credibility to a study when it has been planned based on power to detected specific differences. Briefly mention something, but please do add some words to the statistics section. 

Just a food for thought: were the SCFAs receptors measured in colon tissue? I understand that in liver it might not be highly expressed, but possibly in the colon there might be different, seen the whole pathway interaction between BA, microbiota, and therefore these receptors in the primary organ (colon). 
